# More GP Consultations by Violence Victims: Results from the Representative German DEGS1 Study

**DOI:** 10.3390/ijerph20054646

**Published:** 2023-03-06

**Authors:** Carmen Hunzelar, Yelda Krumpholtz, Robert Schlack, Birgitta Weltermann

**Affiliations:** 1Institute of General Practice and Family Medicine, University Hospital Bonn, Venusberg-Campus 1, 53127 Bonn, Germany; 2Robert-Koch-Institute, Nordufer 20, 13353 Berlin, Germany

**Keywords:** violence experience, general practitioner, population-based study, consultation frequency

## Abstract

Violence is a growing public health problem influencing physical and mental health. Victims tend to contact medical care in the first place, yet a discrepancy between patients’ violence experiences (VE) and general practitioners’ (GP) awareness is reported. The number of GP visits by victims is of interest. Using data of the nationally representative German Health Interview and Examination Survey for Adults (DEGS1), associations between the prevalence of ≥1 recent VE (last 12 months) and the number of GP contacts were analyzed with respect to age, gender, socio-economic status, and health conditions. The DEGS1 dataset comprised persons aged 18 to 64 years (n = 5938). The prevalence of a recent VE was 20.7%. Compared to non-victims, VE victims visited their GP significantly more often in the preceding 12 months (3.47 vs. 2.87, *p* < 0.001), which increased markedly in those who were strongly impaired by a recent physical VE (3.55 GP visits) or psychological VE (4.24). The high frequency of GP contacts in VE victims constitutes opportunities to professionally support this vulnerable patient group and underlines the necessity for GPs to integrate VE as a bio-psycho-social problem in a holistic treatment approach.

## 1. Introduction

Violence is increasingly recognized as a significant public health problem as it has been shown to influence mental and physical health [1,2,3,4]. In 2002, Krug et al. reported on the diversity and worldwide impact of violence categorized into interpersonal violence (e.g., intimate partner violence or armed conflicts), collective (social, political, economic), and self-directed violence (e.g., suicide). In each category, different types of violence, namely, physical, psychological, and sexual violence as well as deprivation can occur [2,5,6,7,8]. Depending on the time in life, duration, and severity of VE, different effects on victims’ health are described [2,3,4,6,9,10,11,12]. Any violence type has been associated with not only short-term but also middle- and long-term adverse health consequences [2,3,5,13,14]. Linkages between violence experiences (VE) and an increased risk of psychological [14,15] and physical chronic diseases [3,7] were shown. Exposure to violence also increased risky health behaviors, e.g., substance abuse or obesity [5,16]. According to a European survey among 42,000 women, one in three women aged 15 years or older have experienced physical and/or sexual violence [17]. While the research focused mainly on the effects of violence on women, especially domestic violence, and intimate partner violence [1,4,8], studies on male VE victims are scarce with inconsistent findings [14]. The current state of knowledge of where violence occurs reveals a gender-specific tendency: men are more likely to experience VE in public spaces and workplaces, and women more in domestic settings [18], which also reflects the motives of perpetration, e.g., in intimate partner violence [19].

A discrepancy between the factual VE cases and legal prosecution is well recognized and attributed to several factors. According to the representative German Victimisation Survey (2017), only approximately one-third of people affected by assaults reported this incident to authorities, mostly because victims think the incident is not serious enough or the police are unable to intervene [20]. In turn, every fifth physically or sexually assaulted woman tended to contact the health care system, especially general practitioners, in the first place rather than seeking help at police departments or social care agencies [21,22,23]. Recent societal developments such as de-stigmatization are leading to an increased consultation of the medical systems by victims of domestic violence, even among males, which fosters a need for professional documentation, forensic examinations, and physician education [22]. Therefore, the World Health Organization (WHO) has developed several clinical guidelines and handbooks to improve health care systems´ response to women subjected to violence [24,25,26], while violence to men or other genders is poorly addressed.

General practitioners (GPs) are considered to have a key role in identifying victims of violence due to injuries caused by violent acts and/or psychological burdens [21,27]. However, a recent study demonstrated a discrepancy between German GPs’ awareness of the prevalence of VE and the German national crime statistics [28]. It concluded that GPs might not seek the necessary dialogue with their patients. Nonetheless, previous examinations illustrated patients‘ readiness to disclose when being asked by their doctor directly [29]. Several influencing factors on GPs’ communication and patients’ comfort to disclose their VE have been examined [30,31]. Furthermore, gender-specific difficulties in help-seeking and disclosure are known, e.g., due to gender norms [32,33,34]. While various psycho-social determinants of health are associated with frequent attendance in GP practices [35], the impact of any kind of VE in women and men on GP contacts is poorly studied.

The representative German Health Interview and Examination Survey for Adults (DEGS1), which is part of the German health monitoring system, provides a broad spectrum of health and health system-use data. It identifies VE by focusing on certain periods of time (recent, in childhood, since the age of 16) rather than surveying a particular type of VE. Although lacking a gender-sensitive approach for VE detection [36], the DEGS1 allows for a better understanding of associations between recent VE and the frequency of GP visits [21,27]. We hypothesized that individuals with recent VE have poorer psycho-physical health and a higher prevalence of GP visits.

## 2. Materials and Methods

This study draws on data from the German Health Interview and Examination Survey for Adults (DEGS1), which is representative of the German population and part of the German health monitoring system. The DEGS1 was carried out by the Robert Koch Institute between 2008 and 2011. Detailed information about the concept and design of DEGS1 has been published [18,37,38,39,40,41]. DEGS1 participants were German citizens aged 18 to 79 years who were surveyed using standardized computer-assisted personal interviews, self-administered questionnaires, standardized examinations, and additional medical tests [18,38,40,41]. The dataset used for this analysis comprised participants aged 18 to 64 years old (n = 5938) because only this age group was asked for experiences of physical and psychological violence [18]. Detailed information on the violence assessment (e.g., ethical aspects and procedural safeguards) can be found in Schlack et al. [18]. Survey-specific weighting factors were used to ensure its representativeness for the German general population [39].

1.DEGS1 measurements of socio-demographic information

Participants had been asked to indicate their age and sex. A multidimensional socio-economic status index (SES) had been calculated and classified into three groups (low, middle, high) using information on education, job, and monthly income [42]. This categorization was based on the international classification Comparative Analyses of Social Mobility in Industrial Nations (CASMIN) [42,43].

2.DEGS1 measurements of participants’ GP contacts and health status

To examine participants´ GP contacts, two variables from the DEGS1 were used: if a participant had a GP, and the number of visits to the GP during the past 12 months. To describe participants´ health, questions on the following topics were analyzed:Subjective health status based on the Minimum European Health Module (MEHM): “What is your health status in general?” (Answer options based on a five-point Likert scale were dichotomized: very good/good health vs. middle/poor/very poor health);The presence of chronic health problems (i.e., long-standing illness with constant treatment and control, e.g., diabetes or heart diseases);The presence of mental health problems, i.e., physician-diagnosed depression ever in life/depression in the last 12 months, and anxiety disorders ever diagnosed in life;The presence of (undiagnosed) current depressive symptoms using the 2-item Patient Health Questionnaire (PHQ-2) [44]. This self-report scale included two items addressing disinterest and depressed mood during the past two weeks on a four-point Likert scale (not at all to nearly every day) [45]. Based on the sum score 0–6, subjects were categorized into two subgroups: no depressive symptoms (0–2) and depressive symptoms (3–6);Data on substance abuse identified risky health behaviors. The level of alcoholic risky consumption (defined as 10 g of pure alcohol for women or 20 g of pure alcohol for men) had been measured by the Alcohol Use Disorder Identification Test—Consumption [AUDIT-C] [46,47] and was categorized into three groups (never drinking, moderate drinking, risk consumption). Participants´ actual smoking behavior was classified into “(occasional) smoker” and “former/never smoker” [48].

3.DEGS1 measurements of self-reported violence

Physical and psychological victimization was assessed in the DEGS1 database using validated items which covered three different periods in life (last 12 months, since age 16, during childhood) [18,49]. For this analysis, two variables were newly composed to investigate if a person had experienced physical and/or psychological violence: (a) ever in life (lifetime VE) and (b) during the last 12 months (recent VE). An overview is shown in Figure 1.

The current state of knowledge on where violence occurs reveals a gender-specific tendency: men experience VE more likely in public spaces and workplaces, and women more in domestic settings [18]. In the DEGS1, victims of recent physical or psychological violence had been asked to indicate not only the responsible perpetrators (partner, family member, colleague, known/unknown person), but also the amount of impairment in well-being by the VE. For the analyses, the impairment in well-being was first trichotomized (not at all; hardly/a little; strong/very strong) and subsequently summarized into “any impairment in well-being due to recent psychological and/or physical VE” (yes, no); non-responders were included in the analyses.

Statistical analyses were conducted using the IBM Statistical Package for Social Sciences (SPSS, Version 25.0) for Windows (IBM Corp., Armonk, NY, USA) with statistical significance set at *p* ≤ 0.05 (two-tailed). All analyses of the DEGS1 data were weighted using the survey-specific weighting factor based on age, gender, region of residence, level of education, community class, and nationality provided by the Robert Koch Institute. Frequency distributions and descriptive estimates were inspected for the entire population. Chi-square tests for categorical data as well as T-tests for numerical data were used to conduct comparisons of the subpopulations of participants with and without VE and VE subcategories (recent and lifetime VE; stratification by kind of violence). A multiple linear regression analysis was used to analyze associations between GP contacts and recent VE. Age, sex, SES, presence of chronic diseases, subjective health status, and physician-diagnosed depression in life were included as covariates. In order to fulfill all statistical requirements and to deal with extreme values, we applied a z-statistic approach and removed cases with z-values greater than +/− 3 as extreme outliers with a respective cut-off of GP contacts >15. In the removed 0.17% of the sample, there was no significant correlation with VE.

## 3. Results

### 3.1. Socio-Demographic Characteristics, Prevalence of VE, and GP Contacts

Among the DEGS1 participants (n = 5938), female and male participants were represented in nearly equal shares. The mean age was 41.6 years, and most participants had a middle SES. Nearly 90% had a GP (n = 5261) whom they attended three times on average during the last 12 months. About 70% had reported a lifetime VE (n = 4042), and about 20% had experienced violence in the past 12 months (n = 1106). A total of 1027 (18.7%) participants had recent psychological violence and 203 (4.8%) recent physical violence experiences, while 124 had experienced both types of violence (2.9%). A total of 933 (86.9%) participants with a recent VE reported impairment in well-being. A strong/very strong impairment was more frequently reported due to psychological than physical VE (42.4%; n = 416; respectively 28.9%; n = 58). Specifically, questions on impairment showed noteworthy missing rates (15.5% missing in impairment by recent psychological VE, 38.0% missing in impairment by recent physical VE), which will be discussed further on. For details see Table 1.

Recent violence was slightly higher in females (90.3%; n = 531 of 594) than males (87.1%; n = 445 of 512). In addition, GP visits in the last 12 months were higher among females than males (3.62 (SD 4.3) vs. 3.24 (SD 5.06)).

### 3.2. Associations between VE and (Mental) Health Problems

The prevalence of a recent VE was higher in the age group 18 to 29 years as well as those with a lower SES (for details, see Table 2). Participants with a recent VE had a significantly higher risk to be diagnosed with depression in life, depression during the last 12 months, and current depressive symptoms. Furthermore, victims had a nearly twice as high prevalence of anxiety disorders compared to those without a recent VE, while the prevalence of having one or more chronic diseases did not differ. Additionally, they were more likely to report a poorer subjective health. Regarding health behavior, participants with a recent VE had higher rates of risky alcohol consumption.

Using the same analytic approach for the subgroup of individuals with and without lifetime VE, similar associations were detected for most mental health parameters and health behaviors. The only differences were that this subpopulation showed a significantly higher prevalence of chronic diseases (25.2%; n = 1058 vs. 21.9%; n = 423, *p* = 0.034) as well as ever in life diagnosed depression (12.2%; n = 516 vs. 8.7%; n = 147) and anxiety disorders (6.2%; n = 247 vs. 3.6%; n = 63). There was no higher prevalence of depression in the last 12 months (51.9%; n = 242 vs. 59.4%; n = 80, n.s.), and no difference regarding their subjective health (20%; n = 842 vs. 21.2%; n = 356, n.s.). As with victims of recent VE, the health behaviors of participants with lifetime VE were found to be risky.

### 3.3. Associations between VE, Impairment by VE, and GP Contacts

Participants with a recent VE reported a significantly higher number of GP visits during the past 12 months than non-victims (mean = 3.47; SD ± 4.98 vs. mean = 2.87; SD ± 4.42, *p* < 0.001). (Figure 2). However, slightly less individuals with recent VE stated having a GP (86.4%; n = 976 vs. 89.7%; n = 4244, *p* = 0.006).

Victims who indicated to be strongly/very strongly impaired in well-being caused by recent psychological violence showed a significantly higher average number of GP contacts than victims who were less impaired (strong/very strongly impaired: mean GP visits: 4.24; SD ± 5.61; hardly/little impaired: 3.08; SD ± 5.18; not impaired: 2.67; SD ± 3.02, *p* = 0.001) (Figure 3). There was no significant difference in GP visits when stratifying by impairment due to physical violence (Figure 3). Concerning lifetime VE, no significant difference was found for having a GP nor for GP contact in the last 12 months.

After specified assumptions (linearity, outliers, multicollinearity, normal distribution of errors, and homoscedasticity) had been tested and were not violated, and after excluding outliers within the number of GP contacts (excluded ≥ 16 contacts during the last 12 months), a multivariate regression model analyzed the association between recent VE and the number of GP contacts based on n = 5229 valid cases (Table 3). Model results (R^2^ = 0.181) revealed that not only did gender, SES, subjective health status, physician-diagnosed depression in life, and presence of chronic diseases show significant influences, but also that victims of the preceding 12 months significantly indicated visiting their GP 0.33 times more often than non-victims.

When repeating the analysis with recent physical and recent psychological VE as separate factors based on n = 5228 valid cases (R^2^ = 0.182), only victims of physical VE significantly indicated visiting their GP 0.69 times more often. Recent psychological VE were no longer significant.

## 4. Discussion

Based on the nationally representative DEGS1 data, this study shows that victims of self-reported violence during the preceding 12 months had significantly more GP contact than those without such experience. The number of GP contacts was highest in individuals who felt strongly impaired by a psychological VE (mean 4.2 GP visits compared to 2.8 in those without VE). The urgent need for adequate professional support is underlined by the fact that VE victims showed a higher prevalence of depressive symptoms, depression, and anxiety disorders.

GPs hold a key position to identify and support VE victims, as they are among the first to be contacted [50]. Therefore, it is reassuring that nearly 90% of VE victims reported to have a GP, and that those with the highest subjective impairment showed the highest number of GP contact, especially those who were psychologically victimized. The medical literature suggests that disclosure and adequate support are not yet necessarily guaranteed despite frequent GP contact. However, from a health service perspective, the high GP attendance of people victimized implies that patients are “at the right place” to receive support if the burden is discussed. Future studies are needed to better understand if communication between GPs and those with recent VE addresses the issue of violence or rather focuses on “general stress”, which may be a subjective relief for the patient but does not lead to violence-specific interventions. This is important as any stressful life event during the preceding 12 months increases the prevalence of GP visits [35]; however, targeted support requires the identification of VE.

Studies show that physician–patient communication on VE is insufficient due to barriers on behalf of both GPs [51,52,53,54] and the patients affected [55]. Zimmermann et al. (2018) revealed discrepancies between GPs´ awareness of their patients´ VE and the German national crime statistics, which can be attributed to several factors. On behalf of physicians, violence is underrepresented in routine medical history-taking, leading to a lack of identification [28]. GPs were shown to be reluctant to inquire about VE for various reasons, e.g., time constraints, the fear of offending the patient, and feeling powerless themselves [51,54]. Additionally, victims´ presentation with unspecific symptoms [56] and/or chronic diseases [2] are believed to contribute to this discrepancy. Additional barriers to disclosure are the fears of not being believed or being judged, self-blame, denial, or even misconceptions regarding GPs´ interest in helping with non-medical issues [55,57]. About three-quarters of female victims would talk about their VE if their doctor directly asked them [29] and would even agree to a routine inquiry by their GP [50,58]. However, there are findings of the under-evaluation of sexual VE by GPs due to, e.g., a lack of counselling skills and specific training [52]. These findings are important as there is a growing openness for professional help on behalf of victims with a growing request to be heard, particularly as societal rethinking is leading towards a growing credibility of at least sexually harassed victims as indicated by the rise of allegations [59].

Repetitively, people of a young age and with low SES were found to be at a heightened risk for recent and lifetime violence [49,60]. The higher attendance of women might be related to more severe VE [61]. Nevertheless, males and females are victims of recent VE [18,49]. The higher GP attendance of females may point towards less help-seeking behavior among male victims. International sociological studies indicate a tendency of female victims to hide their suffered violence due to fear of stigma [62], which might be a reason for male victims as well. The literature suggests a limited readiness for disclosure by men [33], and their tendency to minimize their victimization based on the fear of being stigmatized and losing their reputation [32,63]. To avoid this, routine screening in the health care system regardless of the patient’s gender is essential.

Our findings on the higher prevalence of depression and anxiety disorder are in line with prior studies which showed associations between VE and adverse short-, middle-, as well as long-term health consequences [5,13,64,65,66]. Traumatic experiences are known to be associated with stress reactions [67] which increase the risks for depression and anxiety [68]. In addition, mental disorders are considered a risk factor for VE as well [14,69]. Health problems might not only result directly from violent attacks, but could also constitute a biological response to the strains of victimization, and/or result from maladaptive coping [53]. Therefore, long-lasting physiological mechanisms, i.e., the involvement of the neural, neuroendocrine, and immune system, are implicated in the development of chronic diseases, which could explain why only lifetime VE were associated with the presence of chronic diseases [65,66]. Our finding of a higher prevalence of smoking and risky drinking among VE victims is in line with former studies and may be also understood as victims´ coping strategies [1,16,70].

Strengths and limitations: The measurement of self-reported VE put forth methodical challenges. The DEGS1 study is clearly limited by the lack of data on VE in the elderly, on the severity/dynamics of VE on details regarding the perpetrators, and on gender-sensitive aspects. Furthermore, the DEGS1 was criticized for lacking gender-sensitive aspects. A selection bias cannot be excluded. In addition, a typology of violence including sexual violence is missing in the DEGS1 [36]. As other DEGS1 questions on VE had very high response rates, the high number of missing responses to the questions addressing impairments by VE are not accidental, but rather reflect VE as a socially sensible, supplanted, and in many places tabooed issue. These aspects are well known in VE research with participants´ right of non-disclosure, shame, and recall bias posing special challenges. Aside from methodical challenges in VE measurement, this study does not allow for cause–effect relationships. The DEGS1 is nationally representative for Germany but does not reflect recent events such as the increase in domestic violence during the pandemic [71,72] and traumatized refuges, e.g., from Syria and Ukraine.

## 5. Conclusions

VE need to be perceived as a growing public health problem, particularly because of their detrimental effect on physical and psychological integrity. The high frequency of GP contact by VE victims constitutes an opportunity to provide better care for this vulnerable patient group. Due to recurrent patient contact, GPs have a special role in the detection, prevention, and counseling for victims of violence. Further research is needed to develop and evaluate simple strategies for GP practices to facilitate detection and therapy for VE victims. During the last years, interventions with specific training programs for GPs have been developed to respond effectively to patients affected by VE and to increase clinicians’ preparedness and confidence to meet the needs of these victims [73,74,75,76], yet such programs are lacking in Germany. There is a need for GPs’ awareness of the consequences of VE, both physical and psychological, and the effects that these VE can have on health issues such as depression and anxiety. Our study demonstrated that VE not only influence physical and mental health, but also lead to increased risky health behaviors. Furthermore, GPs should recognize that VE are common among their patients (about one in three) and in all genders, and that patients might have difficulties opening up about their experiences. Therefore, further studies are needed to determine gender-sensitive issues in the disclosure of VE in physician–patient communication.

## Figures and Tables

**Figure 1 ijerph-20-04646-f001:**
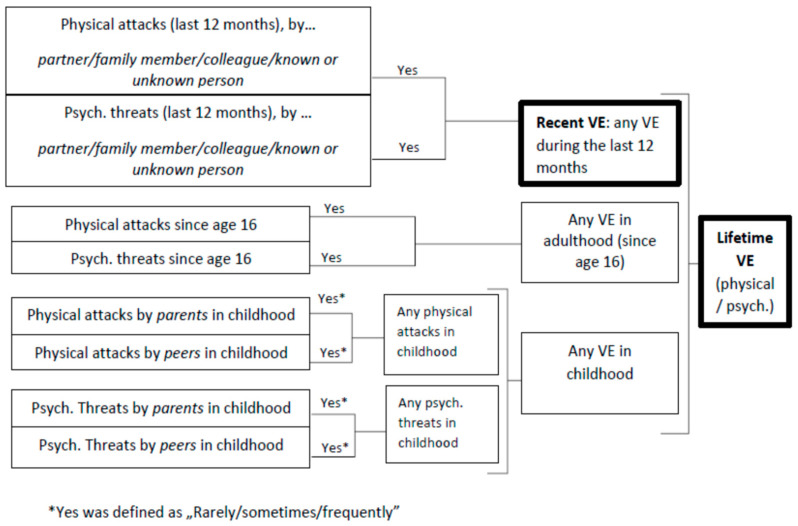
DEGS1 variables computed to measure recent and lifetime physical and psychological violence experience.

**Figure 2 ijerph-20-04646-f002:**
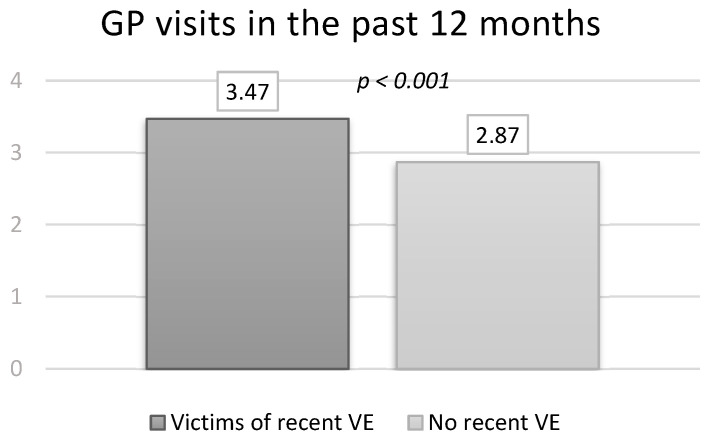
DEGS1 results: GP visits in the past 12 months stratified by recent VE.

**Figure 3 ijerph-20-04646-f003:**
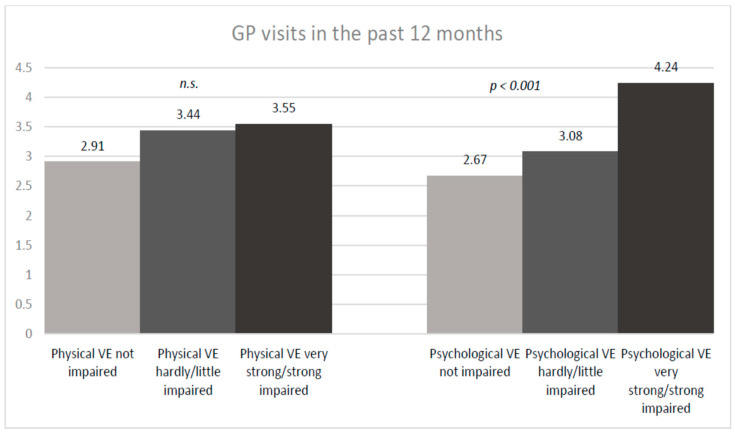
DEGS1 results: GP visits stratified by impairment due to recent physical/psychological VE.

**Table 1 ijerph-20-04646-t001:** DEGS1 participants: socio-demographic characteristics and prevalence of violence experiences (weighted results).

	N * (n = 5938)	% *
Gender (female)	3149	49.4
Age, mean, SD	41.63	13.06
Age groups		
- 18–29	1072	23.2
- 30–44	1730	31.4
- 45–64	3136	45.4
SES		
- Low	847	18.0
- Middle	3521	60.6
- High	1506	21.4
Health and medical information		
Middle to very poor subjective health	1219	20.6
Physician-diagnosed depression (ever in life)	677	11.2
Depression in last 12 months (n = 709)	335	54.4
Current depressive symptoms (PHQ2)	414	7.8
Anxiety disorder (ever diagnosed in life)	316	5.4
Chronic disease	1501	24.4
Health behavior		
(Occasional) Smoker	2115	29.8
Risky alcohol consumption	2517	31.8
Violence experiences (VE)		
Lifetime VE	4042	71.1
- Physical violence	3544	62.5
- Psychological violence	2927	51.7
Recent VE (past 12 months)	1106	20.7
- Psychological violence	1027	18.7
- Physical violence	203	4.8
- Psychological and physical violence	124	2.9
Impaired in well-being due to any recent VE (psychological and/or physical) (N = 949)	933	86.9
- Strong/very strongly impaired by recent psychological violence (n = 981, missing = 15.5%)	416	42.4
- Strong/very strong impairment by recent physical violence - (n = 201, missing = 38.0%)	58	28.9
GP contact		
Has a GP	5261	89.1
GP visits in past 12 months, mean, SD	2.99	4.54

* N (%) unless noted otherwise.

**Table 2 ijerph-20-04646-t002:** DEGS1: Comparison of socio-demographic characteristics, health, and GP attendance of recent victims and non-victims (weighted results).

	Recent VE (n = 1106)		No Recent VE (n = 4700)		*p*-Value
	N *	%	N *	%	
Gender (female)	594	21.1	2486	78.9	n.s.
Age, mean, SD	36.4	13.0	42.9	12.8	<0.001
**Age groups**					
- 18–29	347	32.8	717	67.2	<0.001
- 30–44	337	21.2	1351	78.9	
- 45–64	422	14.0	2632	86.0	
**SES**					
- Low	189	24.3	625	75.7	0.001
- Middle	665	20.9	2812	79.1	
- High	246	16.3	1249	83.7	
**Health and medical information**					
Middle to very poor subjective health	273	23.7	915	19.5	0.010
Physician-diagnosed depression in life	208	18.0	448	9.3	<0.001
Depression in last 12 months (n = 709)	117	62.2	202	49.5	0.025
Current depressive symptoms (PHQ2)	165	15.7	239	5.5	<0.001
Anxiety disorder (ever diagnosed in life)	93	8.7	214	4.5	<0.001
Chronic disease	293	25.9	1176	23.8	n.s.
**Health behavior**					
(Occasional) Smoker	430	43.0	1445	32.3	<0.001
Risky alcohol consumption	400	37.4	1610	34.0	0.026
**Contact to GP**					
Has a GP	976	86.4	4244	89.7	0.006
GP visits in past 12 months, mean, SD	3.47	4.98	2.87	4.42	<0.001

* N (%) unless noted otherwise.

**Table 3 ijerph-20-04646-t003:** DEGS1: Regression model regarding factors influencing GP contact in the last 12 months.

	Estimates	Standard Error	95%-CI Lower Limit	95%-CI Upper Limit	*p*-Value
Constant	2.590	0.182	2.231	2.949	<0.001
Women (ref.: men)	0.352	0.072	0.210	0.493	<0.001
Having recent VE * (ref.: no recent VE *)	0.327	0.123	0.084	0.570	0.009
Middle to very poor subjective health (ref.: very good/good)	1.461	0.165	1.135	1.788	<0.001
Having a chronic disease (ref.: no chronic disease)	1.560	0.126	1.311	1.808	<0.001
Physician-diagnosed depression in life (ref.: never)	0.900	0.197	0.511	1.290	<0.001
Age in years	−0.006	0.003	0.003	0.001	0.086
SES sum score	−0.600	0.011	−0.083	−0.038	<0.001
R^2^ Adj	18.1%				<0.001

* Definition recent VE: physical and/or psychological VE in the last 12 months.

## Data Availability

The DEGS1 dataset underlying this article was provided by the ‘Health Monitoring’ Research Data Centre at the Robert Koch Institute (RKI), which is accredited by the German Data Forum according to uniform and transparent standards (http://www.ratswd.de/en/data-infrastructure/rdc, accessed on 23 February 2023). Data are accessible upon application to interested scientists for anonymous scientific secondary analyses. Detailed information on access, application forms, and guidelines can be obtained from: datennutzung@rki.de.

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
