# Peer review of "More GP Consultations by Violence Victims: Results from the Representative German DEGS1 Study"

_ijerph, 2023, doi:10.3390/ijerph20054646_

Round 1

Reviewer 1 Report

Dear authors,

I congratulate you on the manuscript, which problematizes an important problem of contemporary society. My suggestions for improving the work are as follows: - expand the Introduction and indicate a larger number of references from the domain of secondary literature.

For example, line 29 and 30: "In each category, different types of violence namely physical, psychological, and sexual violence as well as deprivation can occur [5]." This statement needs more than one reference. The whole "Introduction" as well. Once you rewrite "Introduction", you need to rewrite the "Discussion" as well.

Method and Results section are well written.

Please expand "Conclusion". In this form, you have given the essential recommendations, but they need to be supplemented and explained a little more.

Reviewer 2 Report

The paper is of great interest because it presents violence as a universal public health problem of pandemic dimensions. The detection of violence by primary care physicians is a strategy that is being demanded since it is effective for intervention and prevention (studies indicate that victims of intimate violence attend medical consultations 40% more than non-victims violence). Therefore, the contributions of this study are of great interest for research and intervention in this area.

1. Readers are not familiar with the German Health Interview and Ex-11 amination Survey for Adults (DEGS1). Figure 1 shows who perpetrates violence for children under 16 years of age (by parents / by peers). The sample analysed (> 18 years) in this paper does not indicate who perpetrates violence (by partner? / by ...?). Please indicate in the description of the DEGS1, the specific questions about who perpetrated the violence to the participants who reported having suffered it. This question is critical to interpreting the results. Research shows that the effects of violence are different (by partner, by children, by other family members, by work environment, etc.). If the survey does not collect this information, it should be indicated and pointed out as a major limitation, since the usefulness of the results is less, and its contribution loses relevance to the current state of knowledge.  For example, "Nevertheless, males and females are both about equal victims of recent VE [38, 48]" may confuse the reader if it is not specified that the violence suffered by males and females comes mostly from different perpetrators. Likewise, when it is pointed out that "While research focussed mainly on the effects of violence on women, especially domestic violence, and intimate partner violence [1, 4, 17], studies on male VE victims are scarce with inconsistent findings [12]". It is confusing if it is not made clear that the most studied violence in men is that suffered in war, street contexts, etc. because it is the most frequent, and that the most studied violence in women is that suffered in the domestic context because it is the most frequent.

   2. Analysis. The current state of knowledge on interpersonal violence reveals substantial differences between men and women (women experience more violence in intimate settings, men experience more violence in other settings; women tend to underestimate the violence they experience / men tend to underestimate the violence they perpetrate; women tend to perpetrate violence when assaulted / men tend to perpetrate violence without being assaulted; etc... for example Swahn, Alemdar & Whitaker, 2010). See for example ""when it comes to perpetration of IPV, men and women tend to show equivalent rates, yet women are more likely to experience physical injury and to use IPV in self-defense" (Chester & DeWall, 2018, p. 55). These differences make it essential to disaggregate the data by sex, as men's and women's responses to surveys are affected by variables such as those just noted. In the current state of knowledge, it is essential to provide differentiated analyses for male and female victims, since they follow different patterns of behavior. General analyses without this differentiation are a step backwards in the current state of knowledge and lose their usefulness.

 3. Results. Remove the text referring to data already shown in the tables. It is repetitive and unnecessary. Use the text when you want to emphasize the relevance of some data in the tables, but not to repeat the information. Results (sections 3.1., 3.2 and 3.3). Correct numbering (3.1 is repeated in the three results sections). MOST IMPORTANT: Analyze data separately for female and male samples in the three sections.

     4. For multiple regression models (section 3 of the results: renumber 3.1. to 3.3), since the independent variables may have collinearity, a model that allows the INCREMENTAL contribution of each predictor to be assessed (e.g., stepwise) would be more desirable. This is more appropriate and relevant information.

   5. Be careful not to confuse readers. Some explanatory hypotheses may be misleading because they are contradictory to other empirical evidence. For example, "Possible reasons might be men's limited readiness for disclosure [31] and their tendency to minimize their victimization based on the fear to be stigmatized and lose their reputation [30, 64]". International sociological studies indicate that women tend to hide the violence they suffer (international estimates put the number of women who hide domestic violence due to fear or stigma at 80%; 40% lie about the origin of their injuries when they go to hospital emergencies, etc.). The wording of the text leads one to believe that fear or stigma are the reasons for men, or are more frequent in men than in women. Please, qualify this observation appropriately.

   6. Since it uses literature that is difficult to access for the international reader (German language), it is convenient to introduce in the text some explanations on relevant aspects. For example, "Although lacking a gender sensitive approach for VE detection [34], the DEGS1 allows ...". Why does DEGS1 lack gender sensitivity? What criticisms have been made of it, which may be relevant to assess the scope of the results? Please include them in the paper.

Round 2

Reviewer 2 Report

No comments